# Semantic fluency including task switching predicts academic success in medical school

**Sabrina Kaufmann[1], Änne Glass[2], Peter Kropp[3], Brigitte Müller-Hilke**  **[1] \***

**1** Core Facility for Cell Sorting and Cell Analysis, University Medical Center Rostock, Rostock, Germany,
**2** Institute for Biostatistics and Informatics in Medicine, University Medicine of Rostock, Rostock, Germany,
**3** Institute for Medical Psychology and Sociology, University Medical Center Rostock, Rostock, Germany

\* Brigitte.mueller-hilke@med.uni-rostock.de

**Data Availability Statement:** All relevant data are available from the Figshare database under DOI: 10.6084/m9.figshare.12993548.

**Funding:** The author(s) received no specific funding for this work.

## Abstract

### Objectives

The future state treaty on the admission of students to German medical schools calls for a variety of selection criteria among which at least two are required to be independent of school leaving grades. Against this background, the present study investigated achievement motivation and executive functions as predictors of academic success in medical school.

### Material and methods

Second year medical students were assessed for executive functioning by using the Tower of London Test (ToL), a German version of the Controlled Oral Word Association Test (COWAT), the Trail Making Test (TMT-A) and for motivation by using the Achievement Motivation Inventory (AMI). Academic success was evaluated twofold, i) whether the first state exam (M1) was passed at the earliest possible, after completion of the second year and ii) via the grades obtained.

### Results

81 out of 226 students enrolled participated in our study. Passing the M1 was best explained by semantic fluency including task switching. Moreover, academically successful students achieved significantly higher levels in the AMI-facets "compensatory effort" and "engagement". All students scored above average in the TMT-A and average in the ToL.

### Conclusion

Alternating semantic fluency—requiring simultaneously inhibition, updating and task shifting—turned out highly predictive of academic success in medical school. Moreover, higher levels in "compensatory effort" and "engagement" suggested that both, increased energy expenditure as response to fear of failure and elevated readiness to exert effort also impacted positively on success.

**Competing interests:** The authors have declared that no competing interests exist.

## Introduction

Even though school leaving grades show very good prognostic validity for academic success in general and for medical studies in particular [1, 2], they are currently subject to controversial discussions concerning selection criteria for medical school. As opposed to the mere consideration of grade point averages, personality traits and psychosocial skills that may be independent of school grades come to the fore [3]. However, alternative selection criteria need to be weighed against the risk of admitting students with a lower potential for academic performance.

We have previously investigated anxiety as a personality trait and found a negative correlation with academic success. Likewise, we found a negative correlation between anxiety and semantic fluency including task switching [4]. Considering that executive functions allow for mental playing with ideas, taking the time to think before acting, meeting novel and unanticipated challenges, resisting temptations, and staying focused, we were curious whether there are sub-functions of the executive processes that directly impact on academic success in medical school [5]. Executive functions are anatomically and functionally associated with the prefrontal cortex [6] and they represent a heterogeneous construct, for which various theoretical models exist. According to the Supervisory Attentional System model (SAS), there are, on the one hand, automatized sequences of action that are retrieved in routine situations and require no control of consciousness (contention scheduling system). On the other hand, there is a Supervisory Attentional System, that takes control in unexpected situations that require conscious control and adaptation of the current action [7]. In addition, there are working memory models such as Baddeley's and Hitch's, which are based on a central executive function that is controlling selection, maintenance and manipulation of working memory content as well as the two subsystems, phonological loop and visuospatial sketchpad [8]. Finally, Miyake et al. used operationalizations to summarize 3 core components of executive functions, which are i) inhibition of dominant or prepotent responses ("Inhibition"), ii) updating and monitoring of working memory representations ("Updating") and iii) shifting between tasks or mental sets ("Shifting") [9]. These core components are partly independent (diversity) however, they also interact with each other (unity) [9, 10]. For instance, inhibition includes the situational suppression of automatized tendencies, the capacity to realign the behavior accordingly, and the ability to consciously focus attention on a specific goal while suppressing internal and external distractions [5]. However, efficient inhibition also requires sufficient working memory and updating which in turn implies storing information and working with it [10–12]. In detail, working memory is key for understanding test instructions and applying them, but also for monitoring their execution and adjusting the behavior in case of errors. Monitoring is thus required to ensure that a test assignment is executed as accurately as possible and is particularly demanding under time-constraints. Building on the three core components of inhibition, updating and shifting are more complex functions like planning and problem solving which includes the capacity to plan strategically and in a goal oriented manner to avoid mistakes [5]. Executive functioning and in particular working memory have repeatedly been associated with scholarly success in children and adolescents however, little is known about medical students [13, 14]. We therefore used neurophysiological tests to discriminate if the capacity to inhibit, update, shift tasks or to plan and solve problems were positively related to academic achievements.

Beyond executive functions, we were also interested whether achievement motivation as a personality trait would impact on academic outcomes. Per definition, achievement motivated behavior includes applying a standard of excellence to evaluate one's actions, and associate the outcomes of those actions with one's own competence [15]. We therefore asked whether

students who set a high-quality standard for their own actions and who at the same time are able to effectively direct their behavior towards a defined goal are academically more successful in medical school.

We evaluated academic success twofold, i) formatively whether students passed the first state exam at the earliest possible—which is after completion of the second year and ii) summatively via the grades obtained in this exam. Indeed, the first state exam is frequently used to confirm school leaving grades as predictors for academic success in medical school [16]. Our study design implied a cross-sectional study and an assessment of medical students during their second year. The tests and inventory used were standardized and validated and the results therefore were compared to standard samples as well as the outcome of the first state exam.

## Material and methods

### Study design and setting

We performed a cross cross-sectional study assessing executive functions and achievement motivation of medical students in their second preclinical year at Rostock University medical school. Recruitment took place from November until December 2017, and individual appointments with the study participants were scheduled between March and May 2018. During these appointments, which lasted between 45 and 60 minutes, achievement motivation and executive functions were recorded. All test procedures were carried out by the same investigator (SK) and in identical test settings. Follow-up lasted until October 2018 and consisted of monitoring the outcome of the first state exam as academic achievement. The study was approved by the local ethics committee (Ethikkommission an der Medizinischen Fakultät der Universitätsmedizin Rostock), it is registered under A 2016–0186 and written consent was obtained from all study participants.

### Participant recruitment

Participants were recruited at the beginning of their second academic year. Detailed information on the scope and intent of the study was given during compulsory group seminars, at the end of which students were invited to participate. Presenting names and email addresses prompted later contact for the appointment of test sessions. Inclusion criteria were a written consent and enrollment in the third preclinical term at the time of recruitment. Test-specific exclusion criteria were color blindness, color ametropia and blindness for the Tower of London test, also blindness for the TMT-A and a lack of command of the German language for the verbal fluency test. 81 out of 226 students enrolled in this particular academic year participated in our study which corresponded to a response rate of 35.8%. The mean age of our study participants was 21.06 (± 2.26) years, 25.9% (n = 21) were male and 74.1% (n = 60) were female. This proportion of females turned out significantly higher than 59.3% (86/145) in the rest of the academic year (p = 0.0298).

### Outcome measures

We assessed executive functions and achievement motivation using validated tools (see below). Academic achievements were assessed twofold, formatively and summatively via the outcome of the first state exam (M1). German medical students sit their first state exam (M1) after successful completion of the first two preclinical years. Successful completion is defined as having passed all local exams that are mandatory within this time frame. Failure in a single exam prevents admission to the M1 state exam. For our data analysis, we classified all study participants with reference to their M1 state exam into "passed" for the academically successful

students and "not admitted / failed" for the less successful ones (formative assessment). For summative assessments, we collected the individual grades achieved in the exam.

## Data collection tools

Achievement motivation was assessed via the short version of the German Achievement Motivation Inventory (AMI, LMI-K in German) [17]. This inventory consists of 30 items related to professional success and assesses the following facets: goal setting, persistence, flexibility, internality, compensatory effort, eagerness to learn, competitiveness, dominance, engagement, flow, pride in productivity, status orientation, confidence in success, and preference for difficult tasks. Items were evaluated on a 7 point Likert-scale ranging from "does not apply" to "applies completely". Evaluation included the calculation of an overall sum of values representing a global measure of achievement motivation, as well as the determination of facet-specific values to represent the individual facets of achievement motivation.

As inhibition, updating and shifting are partly interdependent, they can hardly be extracted by a single test procedure [9, 10]. However, by combining tests that predominantly assess updating (Trail Making Test) [17, 18], inhibition and updating (word fluency test without task shifting), all three core components (word fluency test with task switching) [19–21], or planning and problem solving (Tower of London test) [22, 23], we aimed to find out if a single one of these executive subfunctions suffices as prerequisite for academic success in medical school.

The part A of the Trail Making Test measures visomotoric speed and at the same time requires constant monitoring of the execution and appropriate error correction, as the respective matrix is not included in the evaluation if a given number of errors is exceeded [17, 18]. The participants needed to connect in a paper and pencil format the numbers one to ninety as fast as possible. Their task was to finish four different number-matrices and evaluation was based on the mean processing time per matrix.

The German version of a Controlled Oral Word Association Test assesses verbal fluency-either as phonemic (letter) or semantic (category) fluency and either includes task switching or does not [19]. The COWAT therefore measures inhibition and updating in the absence of task switching and it measures inhibition, updating and cognitive flexibility when task switching is included [20, 21]. In four different subtests, the participants needed to generate within two minutes each as many different words as possible. For analysis, the numbers of correct words were counted, errors and repetitions were excluded.

The Tower of London Test (ToL) predominantly records planning and problem solving [22, 23]. In our setting, the participants had to solve 20 problems of varying difficulty levels by moving three differently colored wooden balls as quickly as possible from a given starting to a predetermined target position—vertical bars of different length—in a defined number of moves. The subsequent evaluation was based on the number of solved problems by using the given number of moves.

All relevant data are available from the Figshare database under DOI: 10.6084/m9.figshare.12993548.

## Data analysis

All results obtained were analyzed using SPSS Statistics for Windows, version 25 (IBM Corp.). First, descriptive statistics (medians, interquartile ranges, frequencies) were performed to present response rates, sample-characteristics and exam results. Fisher's exact test was used to compare the proportions of male and female students in our cohort to the rest of the academic year. Furthermore, the results of the questionnaires were transformed into average percentile rank values and compared with those of the standard samples cited in the test manuals. To

assess the influence of 7 potential predictors (executive functions and achievement motivation) on passing the M1 in time, we performed a binary logistic regression analysis in a multiple approach, controlling for age and gender. The adjusted odds ratios ($OR_{adj}$) with their respective 95% confidence intervals (CI) and P-values are given here. The cut-off for variable selection was set to .20.

To investigate mean differences between academically successful and less successful students, we applied the t-test for the normally distributed data or the Mann-Whitney U-test otherwise, each for independent samples. Correlations between M1 results (grades) and facets of motivation were tested via Spearman Correlation analyses. Data were tested for Gaussian distribution using the Shapiro-Wilk test. P-values <0.05 were considered statistically significant.

## Results

### Our study cohort was dominated by female students

Out of 81 participants, 51 passed the M1 state exam at the end of their second academic year. Of these, three obtained the grade "1" which is best, 23 obtained grade "2", 23 obtained grade "3" and two obtained grade "4". One student obtained grade "5" and thus failed while 29 students did not even obtain admission. For further calculations, we allocated the grade "5" to both groups, those who failed and those who were not even admitted to the M1 state exam. In summary, 41 females and 10 males in our cohort passed the M1 state exam and were therefore defined as academically successful. In contrast, we defined those 19 females and 11 males who did not pass the exam or had not even been admitted as less successful. As shown by the p-values of the adjusted regression analysis in Table 1, neither age nor gender had any influence on passing the M1 exam.

### Semantic fluency including task switching predicts M1 outcome

In order to address if executive sub-functions exist that are exam-relevant, we here employed the Tower of London Test that predominantly assesses the capacity to plan and solve problems, the Trail Making Test that assesses monitoring, and the German version of the Controlled Oral Word Association Test that assesses inhibition, monitoring and cognitive flexibility in case of task switching, respectively. At first, percentile ranks were assigned to the raw values

**Table 1. Semantic fluency including task switching predicts M1 outcome.**

| Potential influences | simple approaches | | | multiple approach | | |
|---|---|---|---|---|---|---|
| | P-value | OR | 95% CI | P-value | $OR_{adj}$ | 95% CI |
| semantic fluency incl. task switching (garments-flowers) [#] | 0.013 | 1.17 | 1.03–1.32 | **0.035** | **1,15** | **1.01–1.30** |
| trail making test [sec] | 0.065 | 0.95 | 0.89–1.00 | 0.299 | 0.97 | 0.91–1.03 |
| sex female vs. male* | 0.095 | 2.37 | 0.86–6.55 | 0.389 | 1.63 | 0.54–4.92 |
| phonemic fluency incl. task switching (G-R) [#] | 0.232 | 1.06 | 0.96–1.10 | - | - | - |
| achievement motivation [scores] | 0.313 | 1.02 | 0.99–1.05 | - | - | - |
| age [years] | 0.386 | 0.92 | 0.75–1.12 | 0.573 | 0.94 | 0.75–1.18 |
| tower of London test [no. of solved problems] | 0.518 | 1.09 | 0.84–1.41 | - | - | - |
| phonemic fluency (K) [#] | 0.791 | 1.01 | 0.93–1.10 | - | - | - |
| semantic fluency (jobs) [#] | 0.798 | 1.01 | 0.94–1.08 | - | - | - |

[#] [numbers of correct words given within two minutes].

* reference category.

and were subsequently compared to the test inherent and age matched standard sample [18, 19, 22, 24].

In the verbal fluency tests, our cohort showed a substandard performance, compared to the standard sample of the manual. As the sole exception, academically successful students ranged slightly above average with respect to the semantic fluency including task switching and they scored significantly better than their less successful fellow students (Table 2). Indeed, regression analysis showed that this subtest is in our setting the one which predicts the outcome of the M1 exam, independently of age and gender: $OR_{adj}$ = 1.15 CI [1.01–1.30]; p = .035 (Table 1). There were no more significant differences between academically successful and less successful students looking at the three other verbal fluency subtests individually (Table 2).

## Medical students scored extremely high for their ability to monitor

With median percentile ranks of 84 and 93% respectively, our participants scored way above the mean value of the standard sample in the Trail Making Test (Table 2). However, the academically successful students did not differ significantly from the less successful ones and indeed, regression analysis did not indicate any influence of the Trail Making Test results on the M1 outcome.

## The capacity for planning and problem solving did not impact on academic success

The capacity to plan and solve problems as assessed via the Tower of London Test ranged slightly below the standard sample among our less successful students and slightly above for the academically successful ones. However, there was no statistically significant difference between both groups of students (Table 2).

## Academically successful students showed significantly higher levels of "engagement" and "compensatory effort"

Achievement motivation was assessed as a global measure by calculating the overall sums of values resulting from the Achievement Motivation Inventory. As for the tests assessing executive functions, we again started out by comparing our cohort to the standard sample of business students provided with the manual [24]. To that extent, the results from our cohort were transformed into percentile ranks and a median of 54% indicated a slightly higher achievement motivation among medical compared to business students. We then compared achievement

**Table 2. Academically successful students score significantly higher in semantic fluency including task switching.**

| Test | not admitted / failed (n = 30) median [25 / 75% quartiles] | passed (n = 51) median [25 / 75% quartiles] | p-value* |
|---|---|---|---|
| phonemic fluency incl. task switching (G-R) | 27 [7.5 / 46] | 36 [12 / 56] | 0.2317 |
| semantic fluency incl. task switching (garments-flowers) | 35 [17 / 56] | 56 [27 / 62] | **0.0202** |
| phonemic fluency (K) | 30 [12 / 38] | 27 [15 / 40] | 0.9218 |
| semantic fluency (jobs) | 47 [13.5 / 65] | 43 [14 / 63] | >0.9999 |
| trail making test | 84 [73 / 98] | 93 [83 / 98] | 0.1775 |
| tower of London test | 44 [35 / 76] | 53 [35 / 76] | 0.5313 |

*resulting from Mann-Whitney Test.

motivation in our cohort with the grades obtained in the M1 state exam yet did not find a significant correlation. Likewise, when comparing academically successful and less successful students for their global achievement measures, there were no differences. However, when analyzing the individual facets of motivation, we found moderate positive correlations between the M1 results and "engagement" (r = 0.2032, CI = -0.3534 to -0.0429, p = 0.0109) on the one hand and between the M1 results and "compensatory effort" (r = 0.2236, CI = -0.3719 to -0.06418, p = 0.0050) on the other. When comparing the academically successful and less successful students for their individual facets of motivation, we found significantly higher levels of "compensatory effort" and "engagement" among the academically successful students (Table 3).

## Discussion

This present study showed that semantic fluency including task switching significantly correlated with passing the first state exam at the earliest possible suggesting, that all three core components of executive functioning are required for scholarly success. The absence of any positive results from the phonemic fluency tests indicated subtle executive differences between semantic or phonemic fluency which await further elucidation. The absence of positive correlations between word fluency without task switching and academic achievements further indicated that the capacity to switch tasks is indispensable for academic success. Likewise, an absence of positive correlations between the Trail Making Test results and academic achievements suggested that working memory by itself did not suffice to be academically successful, neither did the capacity to plan and solve problems.

As a note of caution, verbal fluency tests not only measure executive control ability like inhibition, updating and task switching, but also verbal ability. However, as our study cohort included excellent and well educated students only, we believe that they had comparable vocabularies at their disposal and we therefore rule out that we measured differently sized vocabularies, only. Instead, we believe that task switching allowed for the differentiation between academically successful and less successful students. While the capacity to inhibit

**Table 3. Academically successful students showed significantly higher levels of "engagement" and "compensatory effort".**

|  | no. of items | M1 not admitted/failed | | M1 passed | |  |
|---|---|---|---|---|---|---|
|  |  | median | IQR | median | IQR | p-value |
| Facet of motivation: |  |  |  |  |  |  |
| goal setting | 1 | 6.00 | 1.00 | 6.00 | 2.00 | 0.996 |
| persistence | 1 | 6.00 | 1.25 | 6.00 | 1.00 | 0.877 |
| flexibility | 1 | 6.00 | 1.00 | 5.00 | 1.00 | 0.647 |
| internality | 1 | 5.00 | 2.25 | 5.00 | 2.00 | 0.980 |
| compensatory effort | 1 | 5.00 | 2.00 | 6.00 | 1.00 | **0.031** |
| eagerness to learn | 1 | 5.00 | 2.00 | 5.00 | 2.00 | 0.378 |
| competitiveness | 1 | 3.50 | 4.00 | 4.00 | 4.00 | 0.744 |
| dominance | 2 | 10.00 | 3.50 | 11.00 | 2.00 | 0.215 |
| engagement | 2 | 6.50 | 3.00 | 8.00 | 4.00 | **0.045** |
| flow | 3 | 17.00 | 3.25 | 17.00 | 5.00 | 0.413 |
| pride in productivity | 3 | 17.00 | 2.25 | 18.00 | 2.00 | **0.050** |
| status orientation | 3 | 17.00 | 4.50 | 16.00 | 4.00 | 0.382 |
| confidence in success | 4 | 19.50 | 4.00 | 19.00 | 4.00 | 0.316 |
| preference for difficult tasks | 6 | 30.00 | 7.00 | 29.00 | 6.00 | 0.324 |

IQR: interquartile range.

repetitions and irrelevant responses may be comparable between verbal fluency tests with and without task switching, the instructions to be kept in the working memory are more complex for the former. We therefore favor the idea that coping with a more demanding task demonstrated a more sophisticated working memory and higher cognitive flexibility and was therefore predictive of better academic achievements.

Higher working memory capacity has previously been correlated with better semantic verbal fluency which also supports our finding of semantic and not phonemic verbal fluency predicting academic success [25]. Likewise, higher working memory capacity was associated with a higher capacity to deal with negative emotions [26]. Indeed, we confirmed this finding indirectly by showing that in second year medical students, semantic verbal fluency including task switching correlated negatively with anxiety as a trait [4]. Together with the observation that depression impacts negatively on verbal fluency [27], these results call for further studies investigating the connectedness between the capacity to regulate emotions and academic success in medical school.

As for achievement motivation, the overall AMI score in our cohort neither correlated with the results of the first state exam, nor did it predict whether a student would pass this exam at the earliest possible, after completion of the second year. A possible explanation could be a ceiling effect with regard to the overall achievement motivation in our cohort. It remains a possibility that an above-average achievement motivation among medical students requires the long version of the Achievement Motivation Inventory in order to detect narrow differences. However, the differentiated examination of the individual facets of the short AMI facets revealed a statistically significant correlation between "engagement" and "compensatory effort" and the M1 results on the one hand and a significant difference between academically successful and less successful students with respect to these facets on the other. Even though we are aware that a differentiated examination of the individual AMI facets needs to be conducted cautiously, we would like to interpret our results to the effect that an increased willingness for effort in the form of "engagement" may be predictive of academic success in medical school as was previously suggested for health science students [28]. Likewise, we believe that the "compensatory effort" characterized as increased vigor to respond to fear of failure also impacts positively on academic success. Indeed, the risk preference model differentiates between "achievement of success" and "avoidance of failure" as motivational systems and it seemed that the latter is supportive of better performance in tough tasks [29–32]. The impact of the individual AMI facets on success in medical school needs to be investigated in more detail, not only by inclusion of larger cohorts but also by using the AMI long version which elaborates all 17 facets of achievement motivation via 10 items each.

Our study cohort included a significantly larger proportion of female students than the rest of the academic year and this finding confirms a previous observation that women are more likely to participate in scientific studies than men [33]. Moreover, the proportion of academically successful students was larger among the female study participants than among the males. However, gender did not show any influence on passing the M1 exam and therefore does not diminish our finding of semantic verbal fluency predicting academic success.

There are limitations to our study, among them the small study cohort of a single academic year sampled at one medical school, only. It remains possible though that results which here appear as trends only—e.g. higher scores in the Tower of London test for academically successful students—become significant in larger cohorts. Also, we are aware that any questionnaire or test harbors the problems of socially desirable answers and trainability, respectively [34, 35]. Indeed, as the Trail Making Test is part of a selection procedure for German medical students, our cohort not only disposes of an excellent working memory, but was most familiar with this test and therefore scored highly above the control cohort.

In summary, executive functions like inhibition, large working memories, cognitive flexibility as well as planning and problem solving are desirable qualities for physicians [36]. In fact, the ability to quickly and efficiently switch between different tasks, particularly in demanding situations with a multitude of competing requirements, is an omnipresent challenge for physicians [37]. We here show, that the combination of inhibition, updating and task switching as measured via semantic verbal fluency predicted academic success already for young students in their early phase at medical school. Semantic verbal fluency including task switching may therefore pose a suitable selection tool for future students. To avoid trainability, parallel test versions need to be developed as these were previously shown to minimize learning effects [35]. Alternatively, traditional questionnaires need to be adapted to situational judgement tests or multiple mini-interviews that pose fair and valid test procedures [38, 39].

## Acknowledgments

The authors would like to thank all study participants who volunteered to contribute to our work.

## Author Contributions

**Conceptualization:** Sabrina Kaufmann, Peter Kropp, Brigitte Müller-Hilke.

**Data curation:** Sabrina Kaufmann, Brigitte Müller-Hilke.

**Formal analysis:** Sabrina Kaufmann, Brigitte Müller-Hilke.

**Methodology:** Änne Glass.

**Project administration:** Brigitte Müller-Hilke.

**Resources:** Brigitte Müller-Hilke.

**Software:** Brigitte Müller-Hilke.

**Supervision:** Brigitte Müller-Hilke.

**Validation:** Sabrina Kaufmann, Änne Glass, Brigitte Müller-Hilke.

**Visualization:** Sabrina Kaufmann, Brigitte Müller-Hilke.

**Writing – original draft:** Sabrina Kaufmann, Brigitte Müller-Hilke.

**Writing – review & editing:** Sabrina Kaufmann, Änne Glass, Peter Kropp, Brigitte Müller-Hilke.

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
