## [Decision Letter · Decision Letter 0]

27 Aug 2020

PONE-D-20-22162

Semantic fluency including task switching predicts academic success in medical school

PLOS ONE

Dear Dr. Müller-Hilke,

Thank you for submitting your manuscript to PLOS ONE. After careful consideration, we feel that it has merit but does not fully meet PLOS ONE’s publication criteria as it currently stands. Therefore, we invite you to submit a revised version of the manuscript that addresses the points raised during the review process.

ACADEMIC EDITOR:  

Please find attached the comments from the reviewer. The reviewer highlights several important issues, which I would like to see addressed. Moreover, it will be important to provide more information about the study design and methods, as in the present form, it may be hard to replicate.

We look forward to receiving your revised manuscript.

Kind regards,

Jorge Leite

Academic Editor

PLOS ONE

Journal Requirements:

"The study was approved by the local ethics committee, is registered under A 2016-0186 and written consent was obtained from all study participants."   

i) Please amend your current ethics statement to include the full name of the ethics committee/institutional review board(s) that approved your specific study.

ii) Once you have amended this/these statement(s) in the Methods section of the manuscript, please add the same text to the “Ethics Statement” field of the submission form (via “Edit Submission”).

Reviewers' comments:

Reviewer's Responses to Questions

**Comments to the Author**

1. Is the manuscript technically sound, and do the data support the conclusions?

Reviewer #1: Partly

2. Has the statistical analysis been performed appropriately and rigorously? 

Reviewer #1: Yes

3. Have the authors made all data underlying the findings in their manuscript fully available?

Reviewer #1: Yes

4. Is the manuscript presented in an intelligible fashion and written in standard English?

Reviewer #1: No

5. Review Comments to the Author

Reviewer #1: Thank you very much for the opportunity to read and review the article entitled: "Semantic fluency including task switching predicts academic success in medical school".

Overall, the article is well-written and easy to follow. I do have several comments and concerns regarding this manuscript.

In the Abstract, the sentence: “Academic success was evaluated twofold, via outcome of the first state exam and whether it was passed in normal time” is not clear enough and needs to be rewritten.

In the Introduction, Page 3, l44, “ourselves” is redundant.

Introduction. The construct of executive function (EF) needs to be better defined and explained. In particular, the authors should present different models of EF.

Next for this particular study, authors need to provide a rationale on why they selected these particular EF as predictors of academic success.

Task description. The authors should provide more details regarding the tasks employed and the EF constructs they are purported to measure. For example, Trail Making Test was used as a test measuring construct of Monitoring. However, TMT is used to measure the construct of visual attention and task switching.

Participants. Please describe the procedures for selecting participants in more details. In particular how many participants, how many (fe)males, mean age and SD. This description of participants should be in this part of the paper and not in the results section.

Figure title is inappropriate (Fig 1. The majority of our cohort was female and academically successful).

Discussion. Some sentences in discussion belong to the Introduction section of the paper, for example, page 14: “Verbal fluency tests are frequently used in neuro-psychological assessments, clinical practice and research and have proven reliable tools to assess executive functions.”

It is very unlikely finding that motivation was not correlated with the results of first state exam. Can the authors offer us an alternative explanation for this finding?

I also think that the results would be more clear if presented in a tabular form instead of figures.

Two suggested references regarding EF for the authors:

Zelazo, P. D., Craik, F. I., & Booth, L. (2004). Executive function across the life span. Acta psychologica, 115(2-3), 167-183.

Best, J. R., & Miller, P. H. (2010). A developmental perspective on executive function. Child development, 81(6), 1641-1660.

Thank you once again for the opportunity to review this interesting study. I wish the authors all the best in their future endeavors.

6. PLOS authors have the option to publish the peer review history of their article (what does this mean?). If published, this will include your full peer review and any attached files.

Reviewer #1: No

---

## [Author Response · Author response to Decision Letter 0]

23 Sep 2020

PONE-D-20-22162

“Semantic fluency including task switching predicts academic success in medical school”

Dear Dr. Leite,

We are grateful for the opportunity to resubmit our revised manuscript and we appreciate both, yours and the reviewer´s comments as they pointed out potential misunderstandings and weaknesses and therefore helped to improve the manuscript.

Please, find below our rebuttal letter that responds to each point raised. The comments are printed in italics, our responses in red. 

We resubmit our manuscript in two versions, i) containing all changes made to the original version marked in red labeled 'Revised Manuscript with Track Changes' and ii) an unmarked version of our revised paper without tracked changes labeled 'Manuscript'.

ACADEMIC EDITOR: 

Please find attached the comments from the reviewer. The reviewer highlights several important issues, which I would like to see addressed. Moreover, it will be important to provide more information about the study design and methods, as in the present form, it may be hard to replicate.

We completely revised our Material & Method section accordingly, please see pages 6-9

 done

2. Thank you for including your ethics statement: "The study was approved by the local ethics committee, is registered under A 2016-0186 and written consent was obtained from all study participants." 

i) Please amend your current ethics statement to include the full name of the ethics committee/institutional review board(s) that approved your specific study.

done

ii) Once you have amended this/these statement(s) in the Methods section of the manuscript, please add the same text to the “Ethics Statement” field of the submission form (via “Edit Submission”).

done

Comments to the Author

Reviewer #1: Thank you very much for the opportunity to read and review the article entitled: "Semantic fluency including task switching predicts academic success in medical school".

Overall, the article is well-written and easy to follow. I do have several comments and concerns regarding this manuscript.

In the Abstract, the sentence: “Academic success was evaluated twofold, via outcome of the first state exam and whether it was passed in normal time” is not clear enough and needs to be rewritten.

Done, please see page 2, L27-28

In the Introduction, Page 3, l44, “ourselves” is redundant.

Agreed – and eliminated

Introduction. The construct of executive function (EF) needs to be better defined and explained. In particular, the authors should present different models of EF.

We revised our introduction substantially, please see pages 3 and 4

Next for this particular study, authors need to provide a rationale on why they selected these particular EF as predictors of academic success.

As we could not find any hard data on EF being associated with academic achievements in medical students, we introduce our rationale in the introduction, page 3 L49-53 and page 4 L80-84. Moreover, we present a rationale for the tests chosen in the M&M section page 8 L153-159

Task description. The authors should provide more details regarding the tasks employed and the EF constructs they are purported to measure. For example, Trail Making Test was used as a test measuring construct of Monitoring. However, TMT is used to measure the construct of visual attention and task switching.

We rewrote the chapter on “data collection tools” and now provide more details on the tasks employed. We also elaborate that we used part A only of the Trail Making Test – thereby eliminating the task switching aspect. We apologize for not pointing that out from the beginning – but are grateful for the chance to clarify this misunderstanding. Please, see page 8/9 L160-179

Participants. Please describe the procedures for selecting participants in more details. In particular how many participants, how many (fe)males, mean age and SD. This description of participants should be in this part of the paper and not in the results section.

Done – please see page 6/7, L119-128

Figure title is inappropriate (Fig 1. The majority of our cohort was female and academically successful).

We concurred with you point below that figures do not necessary help to clarify results – and deleted Figure 1 and presented the results in writing. Please, see page 10, L207-210

Discussion. Some sentences in discussion belong to the Introduction section of the paper, for example, page 14: “Verbal fluency tests are frequently used in neuro-psychological assessments, clinical practice and research and have proven reliable tools to assess executive functions.”

Agreed – and deleted from the Discussion

It is very unlikely finding that motivation was not correlated with the results of first state exam. Can the authors offer us an alternative explanation for this finding?

Well – we tried. Please, see page 18, L306-310

I also think that the results would be more clear if presented in a tabular form instead of figures.

We concur with your argument – and introduced Table 2

Two suggested references regarding EF for the authors:

Zelazo, P. D., Craik, F. I., & Booth, L. (2004). Executive function across the life span. Acta psychologica, 115(2-3), 167-183.

Best, J. R., & Miller, P. H. (2010). A developmental perspective on executive function. Child development, 81(6), 1641-1660.

Thank you very much for these suggestions – they are highly appreciated! The latter one is incorporated into our manuscript as reference 10

Thank you once again for the opportunity to review this interesting study. I wish the authors all the best in their future endeavors.

---

## [Editor Report · Decision Letter 1]

10 Dec 2020

Semantic fluency including task switching predicts academic success in medical school

PONE-D-20-22162R1

Dear Dr. Müller-Hilke,

We’re pleased to inform you that your manuscript has been judged scientifically suitable for publication and will be formally accepted for publication once it meets all outstanding technical requirements.

Kind regards,

Jorge Leite, Ph.D.

Academic Editor

PLOS ONE
---

## [Editor Report · Acceptance letter]

14 Dec 2020

PONE-D-20-22162R1 

Semantic fluency including task switching predicts academic success in medical school 

Dear Dr. Müller-Hilke:

I'm pleased to inform you that your manuscript has been deemed suitable for publication in PLOS ONE. Congratulations! Your manuscript is now with our production department. 

Kind regards, 

on behalf of

Dr. Jorge Leite 

Academic Editor

PLOS ONE